# Reverse dynamic nuclear polarisation for indirect detection of nuclear spins close to unpaired electrons

Nino Wili[1], Jan Henrik Ardenkjær-Larsen[2], and Gunnar Jeschke[1]

[1]Department of Chemistry and Applied Biosciences, Laboratory of Physical Chemistry, ETH Zurich, Vladimir-Prelog-Weg 2, 8093 Zurich, Switzerland

[2]Department of Health Technology, Center for Hyperpolarization in Magnetic Resonance, Technical University of Denmark, Building 349, 2800, Kgs Lyngby, Denmark

**Correspondence:** Nino Wili (nino.wili@alumni.ethz.ch)

**Abstract.** Polarisation transfer schemes and indirect detection are central to magnetic resonance. Using the trityl radical OX063 and a pulse electron paramagnetic resonance spectrometer operating in Q-band (35 GHz, 1.2 T), we show here that it is possible to use pulsed dynamic nuclear polarisation (DNP) to transfer polarisation from electrons to protons, *and back*. The latter is achieved by first saturating the electrons and then simply using a reverse DNP step. A variable mixing time between DNP and reverse DNP allows us to investigate the decay of polarisation on protons in the vicinity of the electrons. We qualitatively investigate the influence of solvent deuteration, temperature, and electron concentration. We expect reverse DNP to be useful in the investigation of nuclear spin diffusion and envisage its use in electron-nuclear double resonance (ENDOR) experiments.

## 1 Introduction

Polarisation or coherence transfer schemes are fundamental to modern magnetic resonance (Ernst et al., 1987). Most commonly, these are INEPT in liquid-state NMR, Cross Polarisation (CP) in solid-state NMR, and Dynamic Nuclear Polarisation (DNP) to transfer electron spin polarisation to nuclear spins. The enhanced polarisation leads to an improved signal-to-noise ratio. Additionally, spins with higher gyromagnetic ratio often show faster longitudinal relaxation, such that the necessary relaxation delay between repetitions of the same experiment can be reduced. This further leads to a higher sensitivity per time.

This is only half the story. Rather often in NMR, after an initial transfer from high-$\gamma$ nuclei (usually protons) to heteronuclei, and an evolution period on the latter, the magnetisation is transferred back to the inital spin and then detected. This is referred to as indirect detection. Again, this leads to increased sensitivity, but it also allows to establish correlations if the spectrum of the initial spins is resolved.

DNP is now an established technique for polarisation enhancement of nuclear spins (Ni et al., 2013; Lilly Thankamony et al., 2017). The polarisation of electrons, which are either naturally in the sample of investigation or are added to it, is transferred to nuclei by appropriate microwave irradiation schemes. At high fields ($\gtrsim$ 3.5 T), usually continuous-wave (CW) irradiation provided by gyrotrons is used, and the transfer is quite slow due to the low microwave power available at high frequencies. Recently, several groups introduced schemes using broadband frequency swept excitation (Hovav et al., 2014; Bornet et al., 2014; Kaminker and Han, 2018; Gao et al., 2019; Shimon and Kaminker, 2020). At lower fields and frequencies, already

a quite appreciable number of pulsed DNP variants are available (Henstra et al., 1988; Tan et al., 2019a, c; Redrouthu and Mathies, 2022). A very simple and efficient one is NOVEL (nuclear orientation via electron spin locking), where the electron is spin-locked with a nutation frequency corresponding to the nuclear Zeeman frequency (Henstra et al., 1988). Noteworthy, the electron-nuclear polarisation transfer is achieved without any radio-frequency (rf) irradiation of the nuclei. This is in contrast to CP, where both spins are irradiated with the same nutation frequency. The NOVEL condition is sometimes also referred to as a rotating frame-laboratory frame Hartmann-Hahn matching (Can et al., 2015).

In principle, electron-nuclear polarisation transfer should be possible in both directions. In this work, we show that this is indeed the case, on the example of trityl OX063 in protonated and deuterated solvents, a sample well suited for pulsed DNP (Mathies et al., 2016). After an inital DNP step, the electron spins are saturated. This leads to a situation where the nuclear polarisation is larger than the electron spin polarisation. A second DNP step then causes nuclear-electron polarisation transfer. We refer to this as "reverse DNP". The experiments are performed at 80 K, on a home-built EPR spectrometer based on a fast arbitrary waveform generator (AWG) (Doll and Jeschke, 2017) working in Q-band ($\approx$35 GHz, 1.2 T), corresponding to a proton resonance frequency of about 50 MHz.

A variable waiting time between DNP and reverse DNP allows us to study the decay of nuclear polarisation close to the unpaired electron. As expected, the nuclear polarisation decays much slower than the longitudinal relaxation of the electron spin $T_{1,\mathrm{e}}$. Preliminary results show a profound influence of protonation of the solvent, indicating that spin diffusion away from the paramagnetic centre plays an important role. The proton polarisation decay is enhanced when increasing the trityl concentration from 100 $\mu$M to 5 mM. Finally, periodic inversion of the electron spin also accelerates the decay. We interpret this as an increase in spin diffusion away from the paramagnetic centre due to hyperfine decoupling.

The method holds potential in investigating the influence of spin diffusion away from a paramagnetic centre (Wolfe, 1973; Stern et al., 2021; Tan et al., 2019b; Jain et al., 2021). On the other hand, we envisage the use of reverse DNP in electron-nuclear double resonance (ENDOR) experiments of nuclei with substantial hyperfine couplings (Harmer, 2016; Rizzato et al., 2013).

## 2   NOVEL matching and electron depolarisation

In a first step, we investigate the electron spin depolarisation during DNP using the NOVEL sequence.

There are several ways one can determine the NOVEL condition experimentally. For example, one could perform nutation experiments and then set the microwave amplitude to the desired nutation frequency. In this work, we used a simple spinlock sequence followed by a spin echo as in (van den Heuvel et al., 1992), see Figure 1(a).

The echo intensity as a function of the spinlock strength (with a constant pulse length of 2 $\mu$s) is shown in Figure 1(b). For a vanishing nutation frequency, there is no effective spinlock and accordingly no echo intensity. The intensity is then more or less constant if the nutation frequency is larger than the trityl ESR linewidth ($\approx$12 MHz FWHM at 1.2 T). However, if the NOVEL condition is fulfilled, there is a drop in electron spin echo intensity, because polarisation is transferred to nearby nuclei. In fully protonated solvent, there is only a dip at $\nu_1 = \nu_0(^1\mathrm{H})$. In deuterated solvent, there is an additional dip at $\nu_1 = \nu_0(^2\mathrm{H})$.

The transcription can further be investigated by keeping the spinlock power fixed on the NOVEL condition, and increasing the spinlock pulse length $t_{\mathrm{SL}}$. This is shown for the protonated solvent in Figure 1(c). Note that the sequence was slightly adjusted in this case. At the NOVEL condition, the spinlock power is on the same order as the EPR linewidth, such that transient nutations due to off-resonance effects obscure the electron-nuclear transfer dynamics (see SI). This can be alleviated by starting the spinlock at full power for about 500 ns, at which point the transient nutations have decayed. The spinlock amplitude is then suddenly dropped from maximum power to the NOVEL condition. This corresponds to $t = 0$ in Figure 1(c). The transfer curve shows clear transient behaviour, with a minimum of the electron spin echo intensity at 184 ns. This is consistent with earlier investigations of DNP with trityl radicals (Mathies et al., 2016).

In principle, this transfer could be repeated several times, to accumulate nuclear polarisation. This is usually done for pulse DNP experiments with direct NMR detection. The effect of multiple NOVEL contacts on the depolarisation is further investigated in the SI (section S9), but all further experiments were conducted with single DNP contacts, because the aim was not a simple NMR signal enhancement, but to investigate the spin dynamics.

## 3 Electron saturation and repolarisation by reverse DNP

The flip-flop terms in the effective Hamiltonian during NOVEL matching lead to an oscillation of the *difference* of electron and nuclear polarisation, $P_{\mathrm{E}}$ and $P_{\mathrm{N}}$, respectively. Usually, the electron polarisation is much larger, $|P_{\mathrm{E}}| \gg |P_{\mathrm{N}}|$, such that polarisation is transferred from electrons to nuclei. However, if we saturate the electron spins after a DNP transfer, the nuclear polarisation is larger than the electron spin polarisation $|P_{\mathrm{E}}| < |P_{\mathrm{N}}|$. In this situation, DNP leads to a nuclear-electron polarisation transfer. The pulse sequence for this is shown in Figure 2(a). It starts as before with a NOVEL block, and after a waiting time $T$, the electron spins are saturated. We used a train of small flip angle pulses and delays. A second spinlock fulfilling the NOVEL condition then leads to nuclear-electron transfer. Note that no $\pi/2$ pulse is needed at this point. The electron polarisation builds up along the spinlock axis, and can be read out again by a simple echo. While this detection sub-sequence is formally equivalent to a notched echo (Ponti and Schweiger, 1994), build-up of electron magnetization during the high-turning angle pulse differs.

Figure 2(b) shows the echo intensity as a function of the nutation frequency of the *first* spinlock, while the power of the second spinlock was fixed at the optimum (i.e. the minimum determined in Figure 1(b)). Clearly, the signal is highest if the spinlock nutation frequency matches the proton Larmor frequency. Interestingly, a signal can be recovered even if $T$ is set to 20 ms (red line), which corresponds to $T > 5 \cdot T_{1,e}$. Note that a $+/-$ phase cycle was used on the very first $\pi/2$ pulse and the detection phase. This proves that there is a correlation between the first NOVEL block and the detected signal, even if $T \gg T_{1,e}$. These findings indicate that nuclear polarisation is generated during the first NOVEL block, even without direct proton NMR detection.

As mentioned earlier, the effective Hamiltonian during forward and reverse DNP is the same, and leads to an oscillation of the difference in polarisation. Figure 3(a) compares the *depolarisation* curve (black) of Figure 1(c) with the *re-polarisation* (red). The latter was obtained by fixing the length of the first transfer to the minimum of the depolarisation curve (184 ns) and

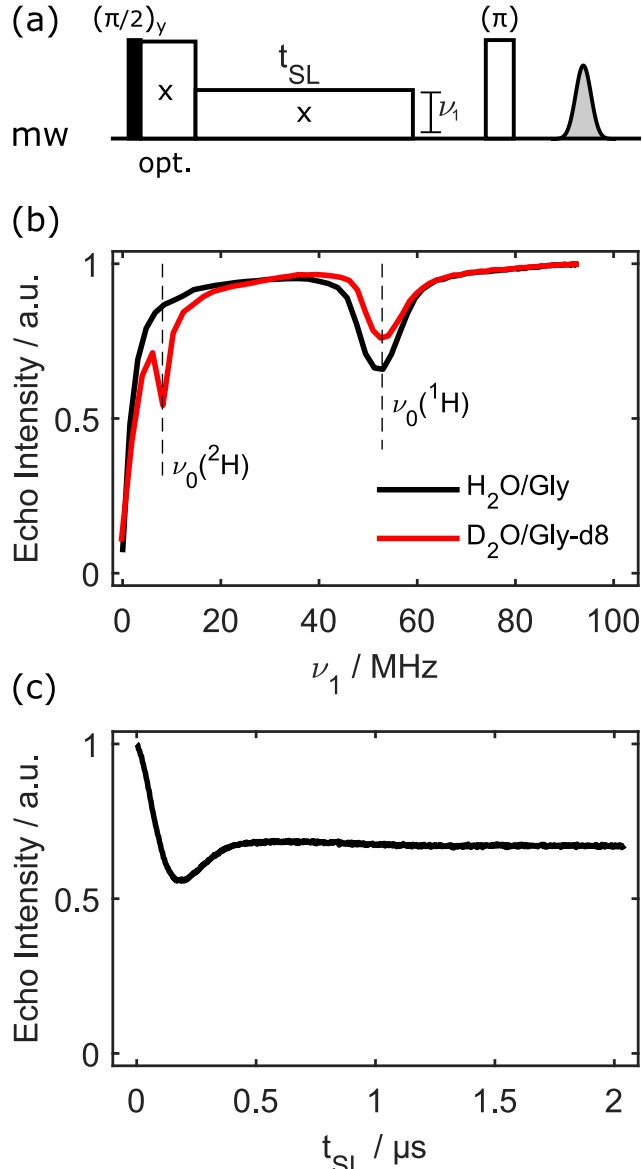

**Figure 1.** Electron spin depolarisation during electron spinlocking. (a) Pulse sequence. The spinlock can optionally start at full power to purge off-resonance effects. (b) Depolarisation power (or nutation frequency) matching with fixed spinlock length $t_{SL}$=2 µs. 100 µM OX063, 80 K (c) Electron depolarisation curve for the sample in protonated solvent, with the microwave power adjusted to the proton Larmor frequency, $\nu_1 = \nu_0(^1\text{H})$, as determined in (b).

varying the length of the second transfer. The power of both transfers was set to the NOVEL condition. The re-polarisation curve is essentially the inverse of the de-polarisation, albeit only a maximum of 1 % signal intensity relative to a Hahn echo could be achieved in this fully protonated sample.

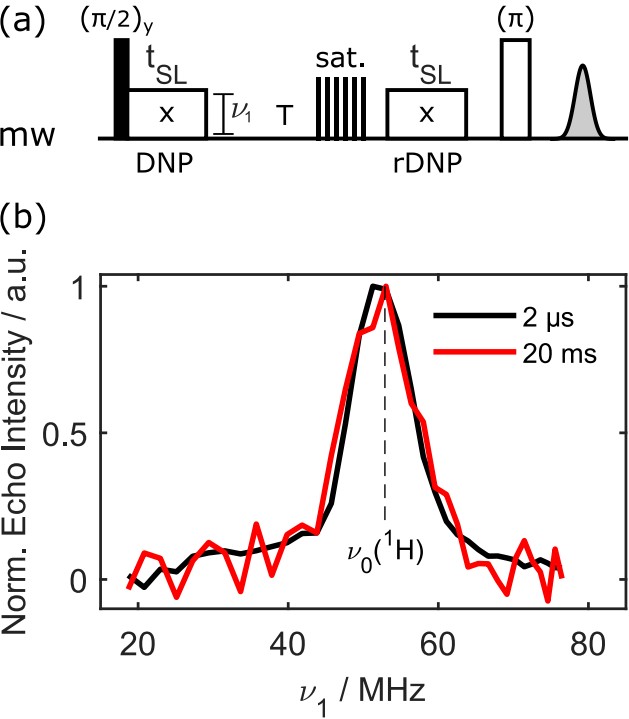

**Figure 2.** Transferring polarisation to nuclei and back to electrons. (a) Pulse sequence. After an initial DNP step and a waiting time $T$, the electron spins are saturated. The next DNP step leads then to nuclear-electron polarisation transfer. The electron polarisation, which builds up along the spinlock axis, can be read out with and echo. Timing details are given in the Materials and methods secion. (b) Repolarisation matching. 100 µM OX063 in protonated water/glycerol, 80 K. The microwave power during the first transfer is swept while the second transfer is kept the same with optimised parameters. Different colours indicate different values of $T$.

This efficiency is quite poor. We assume that multi-spin effects involving several nuclei play a role in this (Henstra and Wenckebach, 2008). Additionally, the nutation frequency of the spinlock is inhomogeneously broadened because the mi-
crowave power is different in different positions inside the resonator (as visible in 2(b), and also in nutation spectra in the SI).

A simple way to improve the robustness of NOVEL with respect to microwave inhomogeneity is to use ramped-amplitude (RA) NOVEL (Can et al., 2017). In this case, the polarisation is transferred adiabatically, analogous to ramped-amplitude CP (Hediger et al., 1994), which improves robustness and potentially increases the maximal polarisation that can be transferred
(details about RA-NOVEL can be found in the SI.) Figure 3(b) shows the re-polarisation curves using RA-NOVEL both in fully protonated (black) and fully deuterated (red) solvent. The first polarisation step was optimised for both solvents individually, also using RA-NOVEL. In the case of fully deuterated solvent, relative echo intensities of 10 % could be achieved after the two transfer steps. This is already much more promising for future uses. Note that even in deuterated solvent, there are still 48

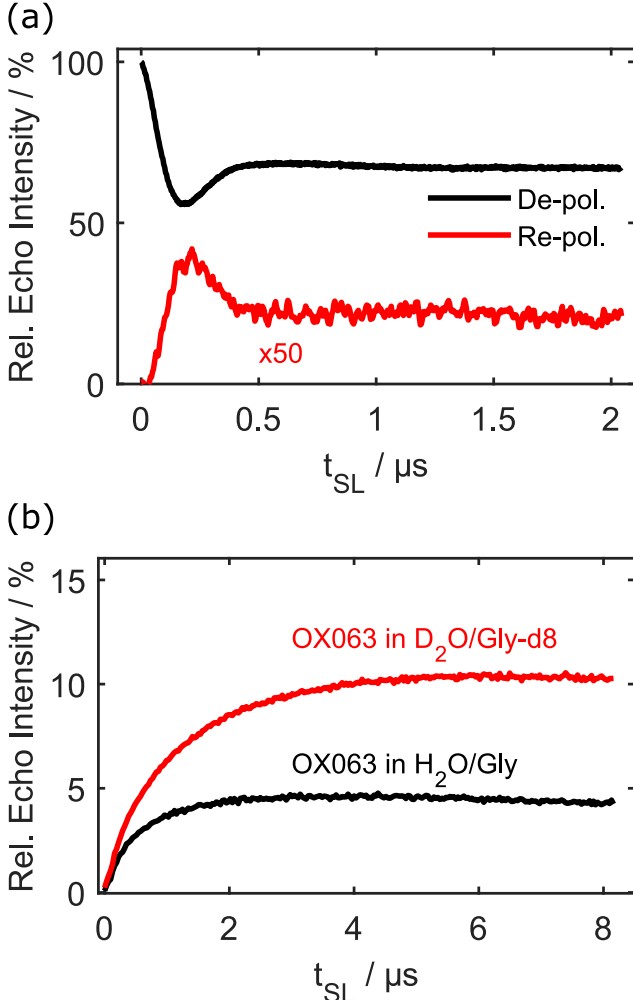

**Figure 3.** Repolarisation dynamics. (a) Comparison of the depolarisation curve in Figure 1(c) with the corresponding repolarisation curve. The latter was measured by sweeping the length of the second DNP step while keeping the first transfer fixed with optimised parameters. 100 μM OX063 at 80 K. (b) Repolarisation curves using ramped-amplitude (RA)-NOVEL in different solvents. In both instances, the protons were polarised and depolarised. All data measured with 100 μM OX063 at 80 K.

non-exchangeable protons in OX063. We speculate that the transfer efficiency could be improved for less abundant nuclei, but experiments are needed to test this hypothesis.

## 4 Proton polarisation decay

We now turn our attention to the decay of the nuclear polarisation during the waiting time $T$ in Figure 2(a).

As a comparison and benchmark, we first measured the longitudinal electron spin relaxation time $T_{1,e}$, at temperatures of 50 K and 80 K. The inversion recovery curves and best fits (single exponential) are shown in Figure 4(a). As expected from results in the literature (Chen et al., 2016), there is a strong temperature dependence of $T_{1,e}$.

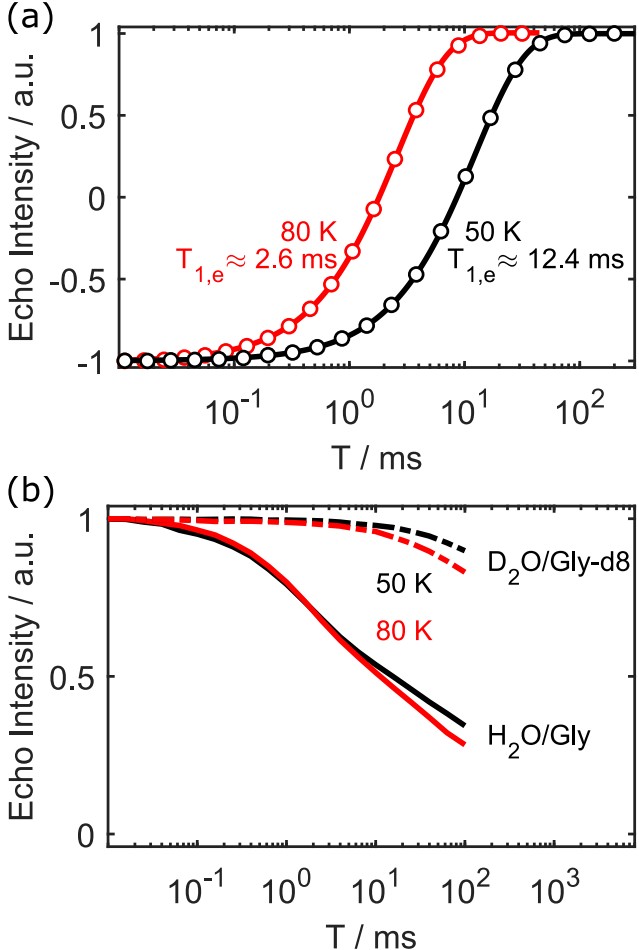

**Figure 4.** Electron inversion recovery (a), and decay of nuclear polarisation (b) at 50 K and 80 K. 100 μM OX063. $T_{1,e}$ is unaffected by deuteration of the solvent.

We then measured the decay of proton polarisation during the interval $T$ for 100 μM OX063 in both protonated and fully deuterated solvent. The results are shown in Figure 4(b). We only measured up to 100 ms, due to software and AWG-memory constraints. These constraints did not pose limitations in any of our previous work, because repetition times in EPR are usually one to two orders of magnitude shorter. We expect to solve this problem in the near future. Even without full characterisation of the decay curve, some qualitative results can still be deduced. First, the proton polarisation decay is much slower than $T_{1,e}$, even in fully protonated solvent. Second, there is only a weak temperature dependence of the proton polarisation decay between 50 K and 80 K. Lastly, there is a very pronounced difference between deuterated and protonated solvent. In the deuterated case,

there is still 90% of the polarisation left after 100 ms. Note that simple (stretched) exponential functions did not give satisfying fits to the experimental data. While sums of stretched exponentials might work, we would like to refrain from naïve fittings in the absence of complete experimental data (i.e. decayed to zero) and adequate quantitative models. Such a model might look similar to (Stern et al., 2021), and could be the topic of future work. In the context of this work, the clear qualitative differences suffice to illustrate that our method can be used to characterise the polarisation dynamics of protons close to the paramagnetic centre. We tentatively assign the faster proton polarisation decay in protonated solvent to increased nuclear spin diffusion away from the unpaired electron in protonated solvent.

Typical DNP measurements are conducted with substantially higher electron spin concentrations than what we used in the experiments so far (100 µM). We thus tested the influence of much higher concentrations, i.e. 5 mM, see Figure 5(a). Clearly, the proton polarisation decay is accelerated at higher electron spin concentrations. Additionally, the slight temperature dependence between 50 K and 80 K vanishes at these elevated concentrations. Note that $T_{1,e}$ is unchanged between 100 µM and 5 mM (see SI). However, we would like to point out that the broadband chirp pulses used for the inversion recovery measurements are able to invert the complete EPR spectrum, effectively eliminating the influence of electron spin spectral diffusion on the apparent value of $T_{1,e}$. This is not always the case when measuring $T_{1,e}$.

Last but not least, we wanted to test if a periodic inversion of the electron spin during the waiting time $T$ might accelerate the polarisation decay of nearby protons. The sequence is shown in Figure 5(b). Loosely speaking, we hypothesised that this periodic inversion would act as hyperfine decoupling (Jeschke and Schweiger, 1997). Differences of hyperfine couplings (partially) truncate nuclear-nuclear flip-flops, leading to the notion of the "spin diffusion barrier" (Khutsishvili, 1963). Eliminating or reducing the hyperfine couplings should thus increase the spin diffusion rate away from the paramagnetic centre. Indeed, this is what we observed. Figure 5(c) shows the proton polarisation decay without any hyperfine decoupling (solid lines) and with periodic inversion of the electron spins every 30 µs with an adiabatic chirp pulse, both for 100 µM and 5 mM OX063 concentration. In both cases, the periodic inversion increases the decay rate of the proton polarisation. Interestingly, with periodic inversion every 30 µs, there is no difference any more between low and high concentrations. Together, these results suggest that electron-electron interactions influence nuclear spin diffusion away from the paramagnetic centre, as already discussed in (Wolfe, 1973). Note that a decoupling period of $\tau_{dec}$=30 µs is not at all sufficient to completely suppress the hyperfine couplings, which can be in the MHz range. It would be better to invert much more often. However, we did not want to risk harming our microwave amplifier, because its specifications are given for much longer recovery periods. On the other hand we would like to point out that the matrix element in the Hamiltonian that quenches the nuclear spin diffusion is given by the *difference* in hyperfine coupling values of the two nuclear spins, not by the absolute values.

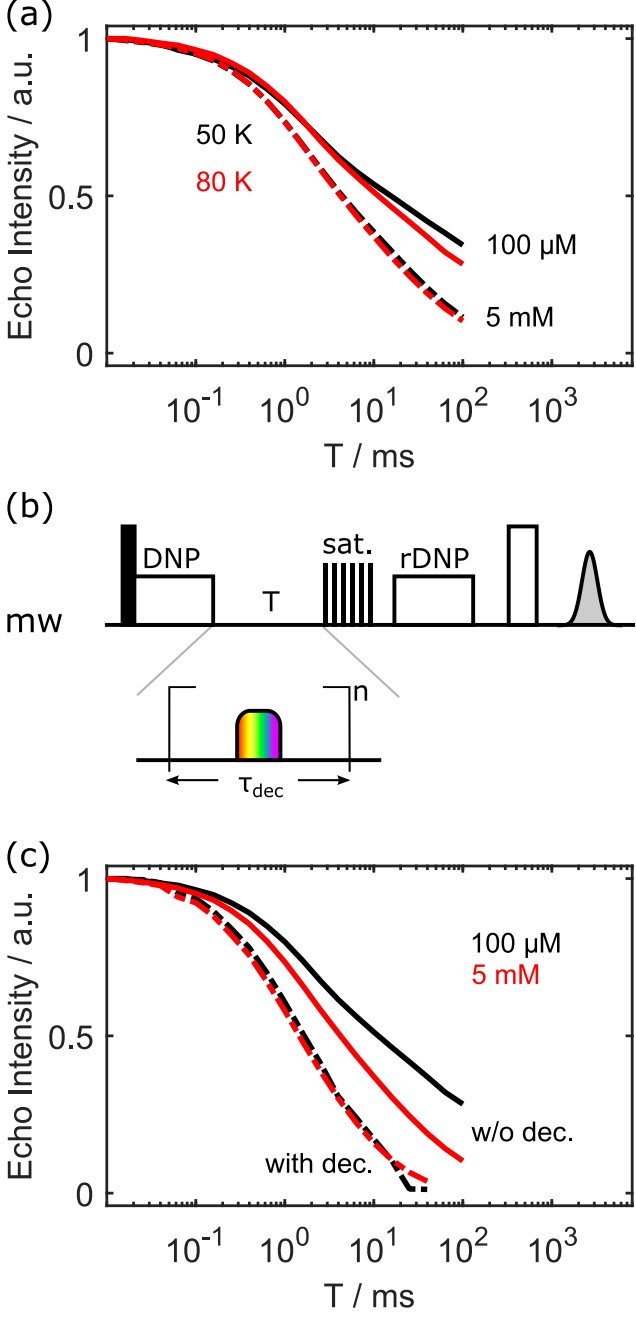

**Figure 5.** (a) Influence of electron spin concentration on the proton polarisation decay. (b) Sequence to investigate the effect of electron decoupling during the waiting time $T$. A chirp pulse inverts the electron spins every $\tau_{\text{dec}}$. This is repeated $n = T/\tau_{\text{dec}}$ times. (c) Effect of electron decoupling. Solid lines: no decoupling was used during time $T$. Dashed lines: The electron spins were inverted every $\tau_{\text{dec}}$=30 μs with an adiabatic chirp pulse. 100 μM OX063 at 80 K.

## 5 Conclusion and outlook

In conclusion, we demonstrated that it is possible to transfer polarisation not only from electron spins to nuclear spins (DNP), but also back (reverse DNP). Using the trityl OX063 in deuterated water/glycerol, an overall efficiency for both transfers of 10% can be achieved with ramped-amplitude NOVEL. The nuclear polarisation is much longer lived than the longitudinal relaxation time of the electrons, $T_{1,e}$. The former lifetime is strongly dependent on deuteration of the solvent and weakly dependent on the electron spin concentration (between 100 μM and 5 mM). Periodic inversion of the electron spins every 30 μs leads to an increased proton polarisation decay, which we tentatively assign to an increased spin diffusion rate away from the paramagnetic centre under hyperfine decoupling.

This work is a proof-of-principle for the feasibility of DNP and reverse DNP for indirect detection of nuclei. We can envisage several ways forward:

The investigation of the influence of different parameters such as temperature, deuteration degree, electron spin concentration, etc. is only qualitative in this work. A systematic screen over a larger parameter range might give very valuable insight into nuclear spin dynamics for nuclei close to the paramagnetic centre. These nuclei are notoriously difficult to access experimentally. One could also combine our approach with standard DNP measurements under the same conditions, with the same parameters for polarisation transfer. Selective inversion experiments using radio-frequency pulses on nuclei between the DNP and reverse DNP might give information about which nuclei actually contribute to the DNP enhancement of the bulk.

Another direction we foresee is to use our indirect detection approach for electron-nuclear double resonance (ENDOR)-type experiments. In the conventional Mims and Davies ENDOR experiments, one generates longitudinal electron-nuclear two-spin order, and not "pure" nuclear polarisation. Another established but less common ENDOR variant, namely cross-polarisation (CP) ENDOR Rizzato et al. (2013), does generate nuclear polarisation, but the read-out is again achieved via longitudinal two-spin order and selective detection of one electron spin transition. It is not obvious how the sensitivity of ENDOR experiments with DNP and reverse DNP will compare to established sequences, this needs to be tested experimentally. We expect at least some advantages, especially in combination with hyperfine decoupling and time-domain ENDOR.

When going to higher fields and frequencies, the NOVEL condition will be difficult or impossible to achieve, because the necessary mw power is not available. In this case, it might be possible to use other pulsed DNP variants, such as electron-nuclear CP (Weis and Griffin, 2006), off-resonance NOVEL (Jain et al., 2017), or the adiabatic solid effect (Tan et al., 2020). All of these have a lower scaling factor (i.e. a slower transfer) than NOVEL, but since the maximum transfer in this work with constant-amplitude NOVEL is already achieved after <200 ns, such a lower scaling factor might still achieve an appropriate amount of polarisation transfer. We expect that further development of modulated sequences such as TOP-DNP (Tan et al., 2019c), XiX-DNP (Redrouthu and Mathies, 2022) or BEAM-DNP (Wili et al., 2022) will also facilitate the use of reverse DNP at higher frequencies, at least in W-band (≈95 GHz). BEAM-DNP was demonstrated for the high-power regime, but would also work at lower power if the modulation frequency was increased — again at the expense of a lower scaling factor.

In this work, we only used a narrow line trityl radical. For radicals with a broader EPR spectrum, such as nitroxides, the bandwidth of the mw sequence will be smaller than the total spectral width. In this case, the experiment should still work

*in principle*. Only a fraction of the electron spins will be excited, leading to orientation selection (Rist and Hyde, 1970), commonly encountered in pulse EPR experiments. In this case, an increased bandwidth of the mw sequence should lead to increased sensitivity.

## 6  Materials and methods

All measurements were conducted on a home-built Q-band (35 GHz, 1.2 T) EPR spectrometer based on a fast AWG and custom-written control software (Doll and Jeschke, 2017). Pulses were amplified using a 150 W travelling wave tube (TWT) amplifier from Applied Systems Engineering. Temperature was controlled with a helium flow cryostat. We used a home-built broadband resonator with high conversion factor, allowing for electron spin nutation frequencies of about 100 MHz (Tschaggelar et al., 2017). OX063 samples were obtained from GE Healthcare. We always used a 1:1 by volume mixture of water and glycerol as solvent, either fully protonated or fully deuterated as indicated in each figure. For each sample, 7 µl were transferred into a 1.6 mm (outer diameter) quartz EPR tube, which was shock frozen in liquid nitrogen and inserted into the cold resonator.

$\pi/2$ and $\pi$ pulses were generally set to 4 and 8 ns. The delay in the echo detection was set to 300 ns. A $\pm$ phase cycle for the very first pulse and the detection was used. Saturation was achieved by a train of 15 pulses of 10 ns duration, spaced by delays of 2 µs. The flip angle of the pulses was about $70\,^\circ$. If not stated otherwise, the time $T$ between the initial DNP step and the saturation was set to 2 µs. The reverse DNP was started 2 µs after the saturation train.

The optional chirp pulses for hyperfine decoupling during time $T$ were of 200 ns length, covered a bandwidth of $\pm 150$ MHz (linear frequency sweep) and were applied at full power (not quite 100 MHz $\nu_1$). The edges of the amplitude modulation were smoothed with a quarter sine wave with a length of 20 ns.

Inversion recovery experiments were measured with the sequence $chirp - T - \pi/2 - \tau - \pi - \tau - echo$. The same chirp pulse as described above was used.

The shot repetition time was usually set to 10 ms for the measurements shown here conducted at 80 K. If the total sequence length was longer than this, the shot repetition time was set to twice the total sequence length plus an additional 20 ms. The number of shots and averages for each figure are given in the SI. Individual averages were saved to check for any systematic saturation behaviour. For the measurements with deuterated solvent, phase-cycling and pre-saturation of the nuclei close to unpaired electrons by multiple electron saturation and reverse DNP was used to mitigate the effects of very slow nuclear relaxation/spin diffusion (see SI).

*Code and data availability.*  All data and code used for this manscript is archived on zenodo: https://doi.org/10.5281/zenodo.6684677

*Author contributions.*  NW designed the research, measured and analysed all data, and wrote the initial draft. GJ and JHAL discussed the results and edited the manuscript.

*Competing interests.*  JHAL is the owner of Polarize ApS that manufactures a dDNP polarizer. NW and GJ declare that they have no conflict of interest.

*Acknowledgements.*  Prof. Matthias Ernst is acknowledged for helpful discussions about spin diffusion measurements in NMR. NW and GJ
acknowledge funding by ETH Zürich grant ETH-48 16-1.

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
