# Peer review of "Reverse dynamic nuclear polarisation for indirect detection of nuclear spins close to unpaired electrons"

_Magnetic Resonance, 2022_

## Author Comment (AC4)

Review in black
Response in blue
Changes in Manuscript in green

Reviewer 1: Daniella Goldfarb

This manuscript presents a new pulse sequence which offers a way to probe the nuclear polarization obtained via DNP by transferring it back to electron polarization after a step of saturation of the electron spins. The authors call this experiment reversed DNP. This is an elegant and original idea and the technique, based on the NOVEL DNP sequence, will be useful in the future to study nuclear spin diffusion and may find applications in ENDOR. The presentation is generally clear (though many experimental details are missing, see below). The authors present a convincing series of experiments, which clearly demonstrate that their idea works and do the right control. I liked the work and I think it fits very well to MR. I recommend publication after addressing the following:

We thank the reviewer for her overall positive comments, and for her suggestions to improve the manuscript.

1. The new method is demonstrated at Q-band frequencies (for the electrons) , ~ 50 MHz for protons, which is way below the standard NMR frequencies at which DNP is applied. This is a general problem of NOVEL. It will be nice to have some outlook regarding possibilities of using this at more relevant frequencies.

   It is indeed the case that the NOVEL condition becomes problematic at higher fields and frequencies. We added a (speculative) paragraph to the discussion and outlook (170-178).

2. The technique is demonstrated on trityl, which at Q-band has a rather narrow EPR signal and with the help of chirp pulses the authors managed to excite the whole spectrum. At a high field the spectrum of trityl broadens and full excitation may not be possible. What will be the consequences of partial excitation (saturation) ? please add to the discussion.

   The chirp pulses were only used as hyperfine-decoupling during the waiting time T. They provide efficient and easy-to-setup inversion. No chirps were used for the DNP steps. We expect that the partial excitation will have the same influence as in other EPR sequences, namely that of orientation selection (i.e. only some orientations will be excited). The experiment should still work in principle. We added the last part to the discussion (180-184).

3. Related to point 2 – will this work for nitroxides, which are often used as polarizers for DNP?

   See point 2.

4. Many experiential details are not given, like length of pulses, delays, properties of the chirp pulses when used, details on the train of saturating pulses, their number and interval , how long it takes to acquire the data. Please give these details in the fig.

captions or in the Experimental part for each figure shown in the manuscript and the SI.

Additional details are now provided in the experimental part, figure captions, and the SI.

5. Figure 5b –it is not clear to me what was the sequence , where are inversion pulses used ? , the sequences show only saturation train.

Since this was also unclear to other reviewers, we added a sequence diagram to Figure 5 showing exactly where the inversion pulses were applied.

6. In the introduction – around line 20, maybe mention that schemes different than just CW irradiation exist.

We added some references mentioning frequency swept excitation both in static and spinning solids (line 22).

7. S9 – please remove the application ruler.

Thank you for pointing this out. We now removed this from the figure.

Reviewer 2: Frédéric Mentink-Vigier

This is an original article by Wili et al. The authors used a coherent DNP pulse sequence to obtain information about the protons surrounding the electron spins. They qualitatively described the effect of deuteration and radical concentration on the decay of the nuclear polarization surrounding the unpaired electrons, as well as used "electron decoupling". The authors prove that the proton polarization decay around the radical is mostly driven by spin diffusion away from the said electron spins. To the best of my knowledge, this is the first time such experiments are carried out. The article is well written and its stimulating content warrants publication in MR. I have minor comments that should be addressed in order to improve the manuscript.

We thank the reviewer for his overall positive comments, and for his suggestions to improve the manuscript.

1. The introduction states that the experiments could be used for ENDOR and to probe the DNP mechanism which I entirely agree with. However, the discussion is rather succinct. How would the reverse DNP be used for ENDOR? My understanding is that the NOVEL condition requires large mw power, as a consequence would it be possible to have "resolution"? Would an ENDOR require a frequency sweep of a rf pulse?

   We would imagine a selective rf pulse between the two DNP steps. The frequency of this rf pulse would be swept from one acquisition to the next, as is the case in Davies or Mims ENDOR. In Davies ENDOR, a selective mw pulse is needed at the start, and thus the power cannot be too high. No such limitation exists in Mims ENDOR, and we also do not expect it in our proposed "DNP-ENDOR". We clarified that a selective rf pulse would be applied between the two DNP steps (line 163).

2. In addition, could we imagine using different pulse sequences? e.g. can we use an eNCP (Rizzato, 2013) to generate the nuclear polarization and then carry out the same reverse CP? If yes, this may be more practical for high field? (Bearing in mind that using offset to have HH condition may be similar as doing off-resonance NOVEL).

   In principle, any DNP step can be used instead of NOVEL. What counts is the transfer efficiency, which has to be high enough than an electron spin echo can still be observed after back-and-forth transfer. This might be less challenging when using concentrations typical for DNP. All of this has to be tested experimentally, and we would hope that other groups will test some of the possibilities.

   We added this point to the discussion, see also answer to reviewer 1, point 1.

3. Could the first step of the nuclear hyperpolarization be replaced by a long ELDOR pulse? Then read the polarization with reverse NOVEL (or other pulse seuqence)? Same question, with DAVIES ENDOR, Iz generated after the rf pi pulse, so DAVIES could also be used?

   See point 2. For the first part of the question. Davies ENDOR could probably be used as a polarizing sequence, although it is very narrow-banded regarding both the

electron as well as the nuclear spectrum. The second part of Davies ENDOR (rf-pi pulse, followed by an electron spin echo) could be used as a read-out sequence, as done in CP-ENDOR.

We think that the changes we incorporated regarding point 2 already imply that any DNP step could – in principle – be used. Since the discussion and outlook already contains a lot of speculation, we decided not to add even more.

4. In general, having more experimental details, such as the delays used, the timing in the pulse sequence figures, would be appreciated. Since MR is a specialized journal, I do not think this would make the manuscript less valuable.

See answer to reviewer 1, point 5. We decided to give more details about timings and pulses in the Materials and methods section, because Figure 2(a) already contains many annotations, and the exact timings are secondary to the understanding of the experiment.

5. "These findings strongly indicate that nuclear polarisation is generated during the first NOVEL block, even without direct proton NMR detection." True but that's like ELDOR, isn't it? So "strongly" is probably not needed.

We removed "strongly" (line 84).

6. Why is the "minimum of the electron spin echo intensity at 184 ns"?

This is due to a dominant dipolar interaction of the electron spin with surrounding nuclear spins, in analogy to Cross Polarisation (although in the latter, one usually observes the target spin, not the initial spin). See also: https://doi.org/10.1016/0009-2614(92)90008-B (already cited, van den Heuvel 1992).

7. "Note that even in deuterated solvent, there are still 48 non-exchangeable protons in OX063. We speculate that the transfer efficiency could be improved for less abundant nuclei, but experiments are needed to test this hypothesis." I am not sure I understand what is meant here. Do you think that a fully deuterated OX063 would enable higher transfer efficiency? If so, could you explain a bit more?

The transfer would probably not be more efficient in a *fully* deuterated OX063, as there would be no close proton that could be polarized. But if there is only a single proton, close to the electron spin, an adiabatic DNP sequence should be able to transfer nearly 100% of polarization to that proton, and accordingly also back. Currently, we only achieve 10% after DNP and r-DNP. It is possible that a lot of signal is simply lost to imperfections in the sequence and relaxation. We measured T1rho, and this is much too long to have an influence. It is not so clear currently what the influence is of the many nuclei. Most likely, the nuclear spins are polarized to a very different extent after the first DNP step. If we could properly invert the complete effective Hamiltonian (as in magic echo experiments, https://doi.org/10.1103/PhysRevLett.25.218), this would not matter, and "all" the polarization could be transferred back to the electron. But in our case, the effective Hamiltonian is the same during DNP and r-DNP, we simply destroy the electron

polarization in between. We plan to perform some multi-spin simulations to explore this in more detail.

Since all of this is rather speculative, we prefer not to expand this in the manuscript and leave the word "speculate" to highlight that it is indeed only speculation.

8. Figure 2, would it be possible to add on the pulse sequence figure the timings in between the pulses? Not as numerical values, but as variables.

See answer to point 4.

9. Figure 3, a, is this deuterated or protonated? In the text it is mentioned but not in the caption. This might improve the readability.

This is for the protonated solvent. We added this in the figure caption.

10. Figure S9 has an issue

See answer to reviewer 1, point 7. We changed the figure.

11. Figure S13, the caption could be improved. I think it would be better if it was described by a separate paragraph.

We moved much of the figure caption into a separate paragraph and expanded the explanation.

Reviewer 3: Marina Bennati

The manuscript by Wili et al describes a new polarization transfer experiment between electron and nuclear spins that consists of two steps. The first step has been reported in the literature and is based on electron-nuclear cross-polarization at the NOVEL condition. The second step is a reverse transfer of polarization from nuclei to electron spins at the same NOVEL condition. The experiment is very interesting as it provides a tool to monitor the dynamics of nuclear polarization after a pulse DNP step, which is otherwise difficult to access. The manuscript is overall clearly written, the experimental effects are robust and seem rationalized. Sometimes details are missing. I would recommend publication in MR after adding the following details.

We thank the reviewer for her overall positive comments, and for her suggestions to improve the manuscript.

1. The first step of the sequence contains a subtle difference to the DNP experiment proposed by the Griffin group, (Can 2015, Mathies 2016). In Figure 2, the NOVEL step is repeated only once, whereas in DNP it is repeated n times (n >~ 1000). In my understanding, this multiple contact is required as one electron spin polarizes a large number of nuclear spins. The question is how much the nuclear spins are polarized in average after only one contact and what is expected then in the reverse polarization step. The authors observe an increase in the intensity of 'repolarized' electron spin in deuterated solvent, indicating that the number of nuclear spins plays indeed a role, see also comment on page 5 ('we speculate that the transfer efficiency could be improved for less abundant nuclei….'). Could the contact time be repeated, which might enhance the efficiency, or are there any hardware constraints to perform this check?

   For this study, we were mostly concerned about nuclear saturation effects that could only show up indirectly when the electrons are detected. That is why we used single contacts only and phase cycling, such that there is no significant nuclear polarization at any given time. This was a conservative choice that facilitated the interpretation of the results, to make sure we understand the spin dynamics as good as possible, and that there is no "history" when performing parameter sweeps/optimisations. Most likely, it would be possible to use several DNP steps before the first reverse DNP step, and then the question is what will lead to the best SNR for a given experiment. We will investigate this further when implementing "DNP ENDOR" experiments.

   We added a small paragraph about multiple DNP contacts (line 64).

   See also the answer to point 2.

2. In context of point (1), in SI 9 the authors report an experiment in which the NOVEL sequence is repeated several times. Perhaps I have missed this, but I could not find a reference to this figure in the main text. This experiment is called 'saturation behavior during depolarization' but it is not entirely clear. Please give the sequence and explain better, also in the main text, what is normalized to 'one' and why the echo is larger than 1 only in the deuterated sample.

We expanded the explanation of section SI 9 (and Fig. S13), and added a reference at the end of section 2 (line 65). It was already mentioned in the Materials and methods section, but only very briefly.

3. Electron polarization decays as a function of waiting times. The authors write (page 6) that the decay is 'supposably' assigned to proton polarization decay. It would be nice to know what other competitive pathways they suspect.

   We used "supposed" in this context with respect to the "proton polarization", because we do no directly detect it. We placed the word in a confusing spot and decided to delete it altogether.

4. Figure 4 shows a clear effect by the solvent deuteration on the 'electron' spin re-polarization dynamics. The effect is tentatively assigned to the faster proton polarization decay in protonated solvent. It is not clear whether we can learn about the nuclear T1n, which would be very important also for ENDOR.

   As mentioned in the text, the decay is neither mono-exponential, nor stretched exponential. We assume that this is the case because different protons at different distances decay with different rate constants, and because spin diffusion is overlayed with "true" proton T1n relaxation. Note that the experiment currently does not directly discriminate between protons with different couplings. We see the sum of signals of whatever protons are polarized (i.e. some coupling is necessary, and probably more strongly coupled protons are polarized more strongly). We expect that rf-frequency selective ENDOR experiments could discriminate between protons with different couplings. In this case it should also be possible to measure T1n as a function of the ENDOR frequency.

5. Effect of 'decoupling sequence' on the polarization decay. The effect of 'decoupling' is interesting but it would be good to see the sequence in Figure 5 and have a rationale of the chosen parameters (length, intensity, repetition rate, setting of adiabatic pulses) also based on the hyperfine couplings (how large are they ?)

   We added a panel to figure 5 clarifying the sequence, and now discuss the settings in the Materials and methods section. They are mostly chosen to properly invert the electron spins, and cautiously not to harm the hardware.

6. Finally, the authors should add some comments about how this experiment could be beneficial for ENDOR, particularly whether it could be extended to radicals with (much) larger hyperfine interaction and the feasibility at higher fields.

   See answers to reviewers 1 and 2.